# Synthesis of Silica-Based Quaternized Adsorption Material and Study on Its Adsorption Behavior for Pu(IV)

**DOI:** 10.3390/molecules27103110

**Published:** 2022-05-12

**Authors:** Zheng Wang, Meichen Liu, Ling Wang, Zhiyuan Chang, Huibo Li

**Affiliations:** Department of Radiochemistry, China Institute of Atomic Energy, P.O. Box 275, Beijing 102413, China; wcyzwz7566395@163.com (Z.W.); liu13069831833@163.com (M.L.); nicekingdom@163.com (L.W.); zychang@ciae.ac.cn (Z.C.)

**Keywords:** silica gel, grafting, reaction mechanism, reprocessing, Pu(IV), adsorption

## Abstract

In this research, we explored the synthesis optimization of the silica-based quaternized adsorption material (SG-VTS-VPQ) and its adsorption behavior for Pu(IV). By optimizing the synthesis process, the grafting amount of 4-vinylpyridine reached 1.25 mmol·g^−1^. According to the analysis of NMR and XPS, the quaternization rate of pyridine groups reached 91.13%. In the adsorption experiments, the thermodynamic experiment results show that the adsorption of Pu(IV) by SG-VTS-VPQ is more in line with the Langmuir adsorption model and the adsorption type is a typical chemical adsorption; the kinetic results show that adsorption process is more in line with the pseudo first-order kinetic model, and the larger specific surface area of SG-VTS-VPQ plays an important role in the adsorption. The results of the adsorption mechanism show that the adsorption of Pu(IV) by SG-VTS-VPQ is mainly complex anion Pu(NO_3_)_6_^2−^ and Pu(NO_3_)_5_^−^. This research provides in-depth and detailed ideas for the surface modification and application of porous silica gel, and at the same time provides a new way to develop the direction of the analysis of pretreatment materials in the spent fuel reprocessing field.

## 1. Introduction

In recent years, people have conducted extensive research on the analysis methods of various impurities in uranium products of nuclear fuel reprocessing. In the reprocessing analysis process, the analysis of key nuclides such as uranium, neptunium, and plutonium is the core of the analysis. Due to the complex reprocessing system, in order to achieve accurate measurement of elements, it is necessary to pre-process the samples before analysis to achieve the separation and purification of the elements to be measured. Therefore, the measurement elements and interference elements can be separated by preprocessing technology, thereby improving the measurement accuracy of the instrument. Plutonium, as one of the impurities in uranium products, is not only an important element for product control, but has an important significance in preventing nuclear proliferation and environmental monitoring [1]. Although the separation and determination of trace amounts of plutonium in simulated uranium products have been reported in the literature, the pretreatment method for analysis of plutonium in reprocessed uranium products is still one of the difficult problems in nuclear industry analysis and detection [2].

In present sample pretreatment work, ion exchange and extraction chromatography are the two most commonly used methods for separating trace amounts of plutonium [3,4]. Due to the advantages of simple equipment, easy control, and high separation efficiency, the ion exchange method has become one of the most commonly used separation techniques. As early as the 1950s, American scientists carried out anion exchange technology to separate plutonium in reprocessing [5,6,7]. The ion exchange resin used in the ion exchange method has the advantages of stable chemical properties and large exchange capacity. However, anion exchange resins have poor rigidity and radiation resistance due to their organic framework, which limits its large-scale application in the field of radioactivity [8]. The extraction chromatography has significant advantages in the separation of trace actinides due to its selectivity of solvent extraction and the high efficiency of chromatography. Thanks to such advantages, different types of extraction chromatography materials have emerged, such as trioctylmethylammonium chloride (TEVA), dipentyl amyl phosphonate (UTEVA), and octyl(phenyl)-N,N-diisobutylcarbamoylmethylphosphine oxide (TRU) extraction resin (produced by TrisKem), which have been widely used in the separation of transuranium elements since their inception [9,10]. Maxwell S et al. used TEVA extraction resin to separate trace amounts of plutonium from uranium matrices, and then used diamide pod ether (DGA) resin for further purification, in which the decontamination factor of uranium in plutonium reached 1 × 10^6^ [11]. Yet the production process of the extraction chromatography materials determines that it has certain defects. The stationary phase of the extraction chromatography materials is easy to lose because it is mostly attached to the surface of the support by coating, which causes its life to be limited and affects the results of analysis [12]. In light of this, it is particularly important to design and synthesize analysis pretreatment materials with good mechanical properties, selectivity, and stability.

In the sample pretreatment process, the carrier of the separation material is the core of the separation process. Therefore, the performance of the carrier material determines the separation effect to a certain extent. Porous silica has become a high-quality carrier for various functional materials, especially adsorption and separation materials, due to excellent physical and chemical properties. Compared with the organic skeleton of ion exchange resin, silica gel has good mechanical properties, acid resistance, and radiation resistance [13,14]. In addition, the surface of silica gel has abundant hydroxyl, and the method of modification with coupling agents can make the functional materials exist on the surface of substrate in a chemically bonded manner, which greatly improves the stability of the functional groups and effectively inhibits the loss of functional materials [15,16,17]. In recent years, separation materials based on silica gel have been extensively studied. Wang et al. prepared a new type of silicon-based functionalized adsorption material silicone esterified adsorbent (SiMaC) by using macroporous silica as the carrier, methacrylic acid as the functional monomer, and ethylene glycol dimethacrylate as the crosslinking agent. At 25 °C, the maximum adsorption capacity of SiMaC for Sr(II) is as high as 142.5 mg·g^−1^ [18]. Chen et al. studied the adsorption behavior of the polyamine- grafted silica composite silicon-based polyamine adsorbent (SAER) on uranium in alkaline solution. This work proved the feasibility of the new silicon-based composite adsorbent to effectively remove uranium from different water matrices [19].

In the authors’ last study, we used coupling agent triethoxyvinylsilane (KH-151) to modify the surface of porous silica gel, explored the mechanism of the alkylation reaction on the surface of silica gel, and optimized the modification process through the grafting rate [20]. On this basis, this study uses 4-vinylpyridine (4-VP) monomer to polymerize and graft on the surface of alkylated silica gel and quaternize it. The prepared silica-based quaternized adsorption material is used for the adsorption behavioral research of trace amounts of Pu. Firstly, SEM, FTIR, NMR, and XPS were used to characterize the materials before and after quaternization to determine the connection mode of organic functional groups on the surface of silica gel and the spatial structure of the synthetic material. Then, the conditions of the grafting reaction were investigated by using the thermogravimetric method (TGA) to determine the optimal synthesis conditions of the silicon-based quaternized material. Secondly, by studying the adsorption behavior of Pu(IV), the adsorption reaction mechanism of the silica-based quaternized adsorption material on Pu(IV) was determined, and the adsorption thermodynamics and kinetics were explored. In summary, due to the limitations of traditional pretreatment methods and the complexity of the analysis process, there are still some deficiencies in the adsorption and separation of trace plutonium in the current reprocessing analysis samples. It is particularly critical to develop a pretreatment material with stable performance, high selectivity, and simple operation. In this study, the surface modification of porous silica gel by chemical grafting combined with the advantages of high selectivity to plutonium of quaternary ammonium salt extractant provides a detailed idea for the synthesis of pretreatment materials with excellent performance and their application, and provides a new direction for material development in reprocessing.

## 2. Material and Methods

### 2.1. Materials

Porous silica gel (pore size about 8 nm, particle size 100–200 mesh) is produced by Kaibang High-Tech Materials Limited Company (Qingdao, China). Xylene (analytical purity), vinyl triethoxysilane (KH-151) (98%), acetonitrile (analytical purity), 4-vinylpyridine (98%, containing inhibitor), azobisisobutyl Nitrile (AIBN, 98%), 1-iodopropane (99%), absolute ethanol, HNO_3_ (guarantee reagent) and NaNO_3_ (99.5%) were purchased from Macleans reagent (Shanghai) limited company. The plutonium and thorium standard solution come from the Department of Radiochemistry, China Institute of Atomic Energy.

### 2.2. Preparation of Silica-Based Quaternized Adsorption Material

The porous silica gel (SG) was modified with the coupling agent vinyl triethoxy silane to obtain alkylated silica gel (SG-VTS), and the reaction conditions for optimal grafting rate have been determined in the previous study [20]. SG-VTS, 4-vinylpyridine (4-VP) and acetonitrile solvent were placed in a DDL-2000 reactor (Eyela, Japan) and stirred for 15 min. Then the initiator azobisisobutyronitrile (AIBN) was added, and the mixture was stirred at a constant temperature for reaction 24 h. After the reaction, the product was washed three times with ethanol, and dried in vacuum at 60 °C for 24 h to obtain the product SG-VTS-VP.

SG-VTS-VP, acetonitrile solvent, and excess 1-iodopropane were placed in the DDL-2000 reactor, and the mixture was stirred at a constant temperature of 80 °C for 24 h to fully quaternize it. After the reaction, the product was washed three times with ethanol, and dried in vacuum at 60 °C for 24 h to obtain the silicon-based quaternized material SG-VTS-VPQ.

### 2.3. Characterization of Silicon-Based Quaternized Material

The S-4800 scanning electron microscopy (Hitachi, Tokyo, Japan) was used to image the microstructures of SG, SG-VTS, SG-VTS-VP, and SG-VTS-VPQ. Fourier transform infrared (FTIR) spectra were obtained on a IS50 FTIR spectrometer (Nicolet, Madison, WI, USA). Nuclear magnetic resonance (NMR) spectra of the SG-VTS-VP and SG-VTS-VPQ were recorded on a Biospin Avance III spectrometer (Bruker, ^13^C, 600 MHz, Boston, MA, USA). Escalab 250Xi X-ray photoelectron spectrometer (XPS) (Thermo Scientific, Waltham, MA, USA) was used to measure surface element content and surface functional groups. SG-VTS and SG-VTS-VP were thermogravimetrically analyzed in DSC3+ thermogravimetric analyzer (Mettler Toledo, Switzerland). The specific surface area and pore structure of SG, SG-VTS, SG-VTS-VP, and SG-VTS-VPQ were measured on an SSA-3000 specific surface area and porosity analyser (Builder, Beijing, China). The specific surface area was calculated by the Brunauer–Emmett–Teller (BET) method. Pore diameter were calculated by Barrett–Joyner–Halenda (BJH) method [21,22].

### 2.4. Calculation Method of Grafting Experiment and Adsorption Experiment

The grafting effect of 4-VP on the surface of silica gel will be judged and optimized by the grafting amount G (mmol·L^−1^). Because 4-VP may form different connection modes during the graft polymerization process, the grafting amount G can be calculated by analyzing the reaction mechanism in the characterization and the thermogravimetric loss of the organic components before and after grafting. For detailed formula, please refer to Equation (10) in the characterization section.

The total concentrations of Pu(IV) were determined by a X fluorescence analyzer (self-developed). The adsorption amount of Pu(IV) in aqueous solution by the SG-VTS-VPQ is calculated by using Equation (1):
(1)$$Q=\frac{\left(C_{0}-C_{e}\right)\times V}{m}$$
where Q (mg·g^−1^) was the amount of adsorption, C_0_ (mg·L^−1^) was the initial concentration of Pu(IV) before adsorption, and C_e_ (mg·L^−1^) was the concentration of Pu(IV) after adsorption. V (L) was the volume of the solution, and m (g) was the mass of the SG-VTS-VPQ.

### 2.5. Optimization Batch Experiments of Graft Polymerization

The single factors of 4-VP dosage, temperature, and initiator concentration were selected to explore the grafting effect.

Group experiments:
(i).Set the 4-VP dosage (molarity): 1 mL (0.31 mol·L^−1^), 2 mL (0.59 mol·L^−1^), 3 mL (0.86 mol·L^−1^), 4 mL (1.11 mol·L^−1^), 5 mL (1.35 mol·L^−1^), 6 mL (1.57 mol·L^−1^), 7 mL (1.78 mol·L^−1^), 8 mL (1.98 mol·L^−1^), 9 mL (2.17 mol·L^−1^), 10 mL (2.35 mol·L^−1^), 11 mL (2.53 mol·L^−1^), 12 mL (2.69 mol·L^−1^), 13 mL (2.85 mol·L^−1^), 14 mL (2.99 mol·L^−1^), 15 mL (3.14 mol·L^−1^). Other conditions: initiator dosage 30 mg, reaction temperature 80 °C.(ii).Set the reaction temperature: 65 °C, 70 °C, 75 °C, 80 °C, 85 °C, 90 °C, 95 °C. Other conditions: initiator dosage 30 mg, 4-VP dosage 7 mL.(iii).Set the initiator AIBN dosage: 1 mg, 2 mg, 3 mg, 4 mg, 5 mg, 6 mg, 7 mg, 8 mg, 9 mg, 10 mg, 11 mg, 12 mg. Other conditions: reaction temperature 85 °C, 4-VP dosage 7 mL.

Orthogonal test:

Orthogonal experiments are designed according to the single factor experimental results, and the design principles are as follows:
(1)Factor selection:

All factors that may have a greater impact on the experimental results need to be tested. Factors with greater influence, universal regularity, and no significant interaction with other factors can be excluded by the control variable method. Under this rule, reaction temperature, 4-VP monomer concentration, and initiator AIBN concentration were determined as orthogonal experimental factors.

(2)Horizontal selection:

For quantitative factors, the selection of levels is more flexible, such as reaction temperature, reagent dosage, etc., usually 2 or 3 levels are selected. In this study, the single-factor experimental results of 4-VP dosage and initiator dosage all have maximum values, so two levels before and after the maximum value should be included in the levels, that is, the number of orthogonal test levels is 3.

Based on principles, L_9_(3^4^) orthogonal experiment was performed with the 4-VP grafting amount as the index to determine the optimum preparation process for the grafting reaction. The levels of orthogonal factors are shown in Table 1.

After each set of reactions, product SG-VTS-VP was subjected to thermogravimetric analysis to determine weight loss. Use Equation (10) to calculate grafting amount.

### 2.6. Adsorption Experiment of Pu(IV) on Silicon-Based Quaternized Material

HNO_3_ and NO_3_^−^ concentration gradient setting experiment:

40 mg·L^−1^ Pu(IV) solution is configured for HNO_3_ and NO_3_^−^ concentration gradient experiment. Set HNO_3_ or NO_3_^−^ concentration: 1 mol·L^−1^, 2 mol·L^−1^, 3 mol·L^−1^, 4 mol·L^−1^, 5 mol·L^−1^, 6 mol·L^−1^, and 7 mol·L^−1^. Other conditions include: temperature 30 °C, adsorption time 1 h. After standing for adsorption, filter, measure the equilibrium concentration of Pu(IV) with X fluorescence analyzer and record it as C_e_. Equation (1) is used to calculate the adsorption capacity.

In order to compare the differences between quaternized silicon-based materials and physical adsorption materials, we use unmodified porous silica materials (pore size about 8nm, particle size about 100–200 mesh) and activated carbon (particle size about 100 mesh) to carry out adsorption experiment of Pu(IV) under different acidity, and other experimental conditions were the same as above.

Kinetic adsorption experiment:

A total of 40 mg·L^−1^ Pu(IV) solution is configured for adsorption kinetics experiments. Set adsorption time: 10 s, 30 s, 50 s, 70 s, 120 s, 180 s, 300 s, 420 s, 600 s, 900 s, 1200 s. Other conditions include: temperature 30 °C, acidity 2 mol·L^−1^ HNO_3_. At each set time, take a small amount of supernatant and filter, measure the concentration of Pu(IV) with X fluorescence analyzer and record it as C_t_. The pseudo first-order, pseudo second-order and Elovich kinetic models were used to determine the rate of the adsorption process.

Pseudo first-order and pseudo second-order kinetics models were given as Equations (2) and (3) [23,24]:
(2)$$q_{t}=q_{e}\left(1-\exp\left(-k_{1}t\right)\right)$$
and
(3)$$q_{t}=\frac{q_{e}^{2}k_{2}t}{1+q_{e}k_{2}t}$$
where q_t_ (mg·g^−1^) is the adsorption capacity at time t (s), and q_e_ (mg·g^−1^) is the equilibrium adsorption capacity. Moreover, k_1_ (s^−1^) was the pseudo first-order kinetic rate constant, and k_2_ (g·s·mg^−1^) was the pseudo second-order kinetic rate constant.

The Elovich kinetic model was given as Equation (4) [25]:
(4)$$q_{t}=\left(1+\beta_{E}\right)\ln\left(1+\alpha_{E}\beta_{E}t\right)$$
where β_E_ (g·mg^−1^) is the desorption constant related to the activation energy of chemisorption and α_E_ (mg·(g·s)^−1^) is the initial adsorption rate.

Adsorption thermodynamic experiment

The following Pu(IV) solutions were configured for adsorption isotherm studies: 5 mg·L^−1^, 10 mg·L^−1^, 20 mg·L^−1^, 30 mg·L^−1^, 40 mg·L^−1^, 50 mg·L^−1^, 60 mg·L^−1^, 80 mg·L^−1^, 100 mg·L^−1^, 120 mg·L^−1^. Other conditions include: temperature 30 °C, acidity 4 mol·L^−1^ HNO_3_, adsorption time 1 h. After adsorption, filter, measure the equilibrium concentration of Pu(IV) with X fluorescence analyzer and record it as C_e_. Equation (1) is used to calculate the adsorption capacity.

Langmuir and Freundlich adsorption isotherm models were given as Equations (5) and (6) [26,27]:
(5)$$q_{e}=\frac{q_{\max}{\mathrm{bC}}_{e}}{1+{\mathrm{bC}}_{e}}$$
and
(6)$$q_{e}=K_{f}C_{e}^{n_{f}}$$
where q_e_ (mg·g^−1^) is the equilibrium adsorption capacity, q_max_ (mg·g^−1^) is the maximum adsorption capacity, C_e_ (mg·g^−1^) is the equilibrium concentration, b is the constant, K_f_ is the Freundlich constant, and n_f_ is the concentration index.

A total of 40 mg·L^−1^ Pu(IV) solution is configured for the adsorption heat experiment. Set adsorption temperature: 20 °C, 30 °C, 40 °C, 50 °C, 60 °C. Other conditions include: acidity 4 mol·L^−1^ HNO_3_, adsorption time 1 h. After adsorption, filter, measure the equilibrium concentration of Pu(IV) with X fluorescence analyzer and record it as C_e_. Equation (1) is used to calculate the adsorption capacity.

The Van’t Hoff equation was used to fit the adsorption capacity at different temperatures to obtain the thermodynamic parameters ΔH and ΔS. Calculate ΔG at different temperatures by the Gibbs equation. Van’t Hoff and Gibbs equations were given as Equations (7) and (8) [28,29]:
(7)$${\mathrm{lnK}}_{c}=-\left(\frac{\Delta H}{R}\right)\frac{1}{T}+\frac{\Delta S}{R}$$
(8)ΔG=ΔH−TΔS
and
(9)$$K_{c}=\frac{C_{s}}{C_{e}}$$
where C_s_ is the concentration of the solid surface at the adsorption equilibrium and C_e_ is the concentration in the solution at the adsorption equilibrium.

Study on the adsorption mechanism of Pu(IV) by SG-VTS-VPQ:

A total of 40 mg·L^−1^ Pu(IV) solution was configured to explore the adsorption mechanism of Pu(IV) on SG-VTS-VPQ. Set adsorbent dosage: 20 mg, 40 mg, 60 mg, 80 mg, 100 mg. Other conditions include: temperature 30 °C, acidity 4 mol·L^−1^ HNO_3_, adsorption time 1 h. After adsorption, filter, measure the equilibrium concentration of Pu(IV) with X fluorescence analyzer and record it as C_e_. Equation (1) is used to calculate the adsorption capacity.

Separation experiment of trace plutonium in uranium matrix:

A certain amount of SG-VTS-VPQ was loaded into a 5-mL extraction chromatography column, and passed through the column with 2 mol·L^−1^ HNO_3_ to make it pre-equilibrated for use. The natural flow rate of the chromatographic column is about 0.5–0.6 mL·min^−1^. Prepare the following U-Pu mixed samples: U concentration 1000 mg·L^−1^, Pu concentration 20 mg·L^−1^, HNO_3_ concentration 2 mol·L^−1^.

Take 1 mL of the above U-Pu sample and pass it through chromatographic column. A total of 14 mL of 2 mol·L^−1^ HNO_3_ was used to elute the uranium, and then 13 mL of 0.3 mol·L^−1^ HNO_3_-0.3 Na_2_C_2_O_4_ was used to elute the plutonium, and the uranium and plutonium eluent were collected separately. The concentrations of uranium and plutonium were measured by X-ray fluorescence analyzer, and the elution curves of uranium and plutonium were prepared. The formula for calculating the decontamination factor of uranium in plutonium is as follows (10):
(10)$$\mathrm{DF}=\frac{U\;\mathrm{content}\;\mathrm{in}\;\mathrm{the}\;\mathrm{sample}/\mathrm{Pu}\;\mathrm{content}\;\mathrm{in}\;\mathrm{the}\;\mathrm{sample}}{U\;\mathrm{content}\;\mathrm{in}\;\mathrm{eluent}/\mathrm{Pu}\;\mathrm{content}\;\mathrm{in}\;\mathrm{eluent}}.$$

Acidity stability test:

A certain amount of pretreated silicon-based quaternary ammonium material SG-VTS-VPQ was placed in 1 mol·L^−1^ HNO_3_, and soaked for 0 h, 4 h, 8 h, 12 h, 16 h, 20 h, and 24 h respectively, and then washed with deionized water until neutral, dried under vacuum at 80 °C and used for the adsorption experiment of Pu(IV). Change the above HNO_3_ concentration to 1 mol·L^−1^, 3 mol·L^−1^, 5 mol·L^−1^, 7 mol·L^−1^, 8 mol·L^−1^, and 9 mol·L^−1^, respectively. After soaking the material for the above time, it was used for adsorption experiments.

A total of 40 mg·L^−1^ of Pu(IV) was prepared for the static adsorption experiments of each of the above soaked materials. Other conditions include: temperature 30 °C, acidity 4 mol·L^−1^ HNO_3_, adsorption time 1 h. After standing for adsorption and filtering, the equilibrium concentration of Pu(VI) was measured by X-ray fluorescence analyzer, which was recorded as C_e_. Equation (1) was used to calculate the adsorption capacity.

Fourier transform infrared spectrometer was used to measure the infrared spectrum of SG-VTS-VPQ after soaking in different acidity for 24 h. The SG-VTS-VPQ and potassium bromide were mixed uniformly in a ratio of 1:100 and placed on a manual tablet machine to make flakes. The test is carried out under the following conditions: the resolution is 4 cm^−1^, the scanning range is 4000–500 cm^−1^, and the scanning is performed 16 times.

Adsorption stability research experiment:

The following Pu(IV) samples were prepared: Pu concentration 20 mg·L^−1^, HNO_3_ concentration 4 mol·L^−1^. First, take 1 mL of the above sample and pass it through the chromatographic column, then wash the column with 15 mL of 4 mol·L^−1^ HNO_3_, and then use 12 mL of 0.3 mol·L^−1^ HNO_3_–0.3 mol·L^−1^ Na_2_C_2_O_4_ to elute Pu, and collect the eluent of Pu. The concentration of the eluent of Pu was measured by X-ray fluorescence analyzer, and the elution curve of Pu was prepared.

The eluted chromatographic column was washed with a large amount of deionized water, and then passed through the column with 4 mol·L^−1^ HNO_3_ to make it pre-equilibrated. The above experiment was repeated 5 times, and the elution curve of Pu was made.

Establishment of Pu standard curve:

Basic information of the self-made X-ray fluorescence analyzer: X-ray tube model Varian OEG-83J; transformer 50 KV, 100 mA; measuring unit PSX-16; controller CTX-16.

Plutonium standard series solutions with mass concentrations of 5 mg·L^−1^, 10 mg·L^−1^, 30 mg·L^−1^, 50 mg·L^−1^, and 70 mg·L^−1^ were prepared with 1 g·L^−1^ of Pu(IV) standard solution. The fluorescence intensity A_Pu_ (au) of the standard series solution was determined by X-ray fluorescence spectroscopy, and the standard curve was drawn as A_Pu_ versus C_Pu_ (Pu concentration).

Measurement conditions: measurement mode: cyclic measurement; number of measurements 3; internal standard Agcom; background subtraction method: direct stripping; XRF measurement time 180s; interval time 180 s.

The standard curve equation for Pu is: y = 67.42x + 20.51, R^2^ = 0.9990. The standard curve is shown in Figure 1, and the linear relationship is good in the mass concentration range of 5–70 mg·L^−1^.

The preparation process of SG-VTS-VPQ and the adsorption process of Pu(IV) are shown in Figure 2.

## 3. Results and Discussion

### 3.1. Characterization of Silicon-Based Quaternized Material

The appearance and SEM images of SG (a), SG-VTS (b), SG-VTS-VP (c), and SG-VTS-VPQ (d) are shown in Figure 3. It can be seen from SEM images that the surfaces of SG (a) and SG-VTS (b) are relatively smooth compared to SG-VTS-VP (c) and SG-VTS-VPQ (d). Moreover, as the grafting process progresses, the appearance of the materials changes significantly. The color of the quaternized SG-VTS-VPQ is yellowish brown, which is similar to pyridine anion exchange resin.

Figure 4 exhibit FTIR of the SG (a), SG-VTS (b), SG-VTS-VP (c) and SG-VTS-VPQ (d). For SG (a) and SG-VTS (b), the changes of surface groups before and after the alkylation reaction have been analyzed in detail in the previous study. For SG-VTS-VP, in Figure 4c, according to the research of Jermakowicz et al., the characteristic peaks at 1419.29 cm^−1^ and 1559.14 cm^−1^ are the C-C stretching vibration in the pyridine ring, and the characteristic peak at 1642.09 cm^−1^ is the C-N stretching vibration in the pyridine ring, which proves that 4-VP monomer has been successfully grafted onto the surface of silica gel. After quaternization, the characteristic peaks of C-C in the pyridine ring almost disappeared because it was covered by the characteristic peak of quaternary ammonium nitrogen at 1472 cm^−1^, which can be clearly observed in Figure 4d [30]. This proves that the pyridine group grafted on the surface of the silica gel has been quaternized.

The ^13^C-NMR spectra of SG-VTS-VP and SG-VTS-VPQ are shown in Figure 5. Figure 5a is the ^13^C-NMR spectrum of SG-VTS-VP, in which 150.3 ppm, 130.7 ppm, and 123 ppm are the absorption peaks of C corresponding to the pyridine ring, 135.5 ppm is the absorption peak of C corresponding to –C≡N in initiator residue groups, and 16.2 ppm is the absorption peak of C corresponding to –CH_3_ in coupling agent and initiator residue groups [31]. In addition, 29.7 ppm, 39.4 ppm, and 58.3 ppm are the absorption peaks corresponding to the secondary, tertiary, and quaternary carbon atoms in the chain respectively [32]. According to the ^13^C-NMR spectrum analysis of SG-VTS-VP, there may be a structure on the surface of the silica gel as shown in Figure 6a after grafting 4-VP.

Figure 5b is the ^13^C-NMR spectrum of SG-VTS-VPQ. Compared with the previous spectrogram, due to the quaternization reaction, the peak positions of the corresponding carbons on the pyridine ring have changed, namely 164 ppm, 145.7 ppm, and 130.1 ppm. After the quaternization reaction, due to the increase in the number of –CH_3_ and –CH_2_– in the structure (the quaternization reagent is 1-iodopropane), the peak positions of the corresponding positions have changed greatly. The strong peak at 11.5 ppm is the corresponding C in –CH_3_; the strong peak at 24.7 ppm is the corresponding C in the remote –CH_2_–. In addition, under the influence of N in pyridine, C in –CH_2_– adjacent to N has a strong absorption peak at 63.2 ppm [33,34]. The peaks corresponding to tertiary carbon and –C≡N remain unchanged. According to the above analysis, after the quaternization reaction, the possible structure of SG-VTS-VPQ is shown in Figure 6b.

The XPS spectrogram of SG, SG-VTS, SG-VTS-VP, and SG-VTS-VPQ are shown in Figure 7. For SG and SG-VTS, comparing Figure 7a,b, Si 2p absorption peak at 103 eV, Si 2s absorption peak at 153 eV, and O 1s absorption peak at 533 eV all exist in both spectra. SG-VTS has a C 1s absorption peak at 285 eV, which indicates that the silane coupling agent is already present on the surface of the silica gel. For SG-VTS-VP, as shown in Figure 7c, after grafting 4-VP monomer, the C 1s peak at 285 eV is significantly enhanced and the absorption peak of N 1s appears at 399 eV, which shows that 4-VP monomer was successfully grafted and polymerized on the surface of the silica gel. However, the spectrum of SG-VTS-VPQ has changed a lot compared with the previous ones, and this change is mainly due to the appearance of the characteristic peak of iodine. As shown in Figure 7d, 47 eV is the characteristic peak of I 4d, 617 eV is the characteristic peak of I 3d5/2, 631 eV is the characteristic peak of I 3d3/2, and 875 eV is the characteristic peak of I 3p [35,36].

In order to further explore the law of the grafting reaction and quaternization reaction of 4-VP on the surface of silica gel, we fitted the N 1s absorption peaks of SG-VTS-VP and SG-VTS-VPQ. For SG-VTS-VP, two deconvoluted peaks centered at 398.9 eV and 398.4 eV were assigned to N-C in pyridine ring and N≡C in initiator residue group respectively [37,38]. The N 1s spectra of SG-VTS-VP was illustrated in Figure 8a. According to the fitting results, the peak area ratios of the two groups are 92.3% and 7.7%, respectively. Because the pyridine ring contains two N-Cs, according to theoretical calculations, the content of N in the pyridine ring is about 85.6% of the total N, and N in the residual initiator group is about 14.4%, namely the amount of 4-VP and initiator residues participating in the reaction accounted for 85.6% and 14.4% respectively.

After the quaternization reaction, the N in the material may have the following four connection modes, which are the N≡C in the initiator residue group, the N-C in the pyridine ring that has not participated in the reaction, the N-C in the pyridine ring that has been quaternized, and the N-C formed by the alkyl group introduced in the quaternization process. Four deconvoluted peaks centered at 398.5 eV, 398.9 eV, 401.3 eV, and 401.7 eV were assigned to the above four connection methods respectively [37,39,40]. The N 1s spectra of SG-VTS-VPQ were illustrated in Figure 8b. According to the fitting results, the peak area ratios of the four groups are 8.8%, 5.6%, 57.5%, and 28.1% respectively.

According to the theoretical calculation based on the fitting result, the number of pyridine groups involved in the quaternization reaction is about 91.1%. The percentage combined with the subsequent thermogravimetric analysis will be used to accurately calculate the grafting amount of 4-VP.

The TG-DTG results of SG-VTS and SG-VTS-VP are shown in Figure 9. In the previous study, we optimized the reaction of the silica gel graft coupling agent KH-151 to obtain the best grafting conditions. Under these conditions, the maximum grafting rate of the coupling agent reached 91.03%, and the weight loss of organic ingredients reaches 13%. The SG-VTS thermogravimetric results are shown in Figure 9a. On this basis, we launched the research of grafting 4-VP. For SG-VTS-VP, as shown in Figure 9b, the range of 30–286 °C is the dissociation process of residual solvents such as acetonitrile and ethanol on the surface of the material, and 286–1000 °C is the dissociation process of organic component grafted on the surface of the silica gel, which includes pyridine groups and initiator residue groups, etc. According to the weight loss and the analysis results in NMR and XPS, the calculation Equation (11) of 4-VP grafting amount as follows:
(11)$$G=\frac{9.8\Delta m-1.3m}{m}$$
where G (mmol·g^−1^) is the grafting amount of 4-VP, Δm is the weight loss of the SG-VTS-VP during the heating process, and m is the weight of the SG-VTS-VP.

Pore size distribution of SG, SG-VTS, SG-VTS-VP, and SG-VTS-VPQ are shown in Figure 10. For SG, the specific surface area and maximum probability pore size calculated by the BET method and the BJH method are 540.9 m^2^·g^−1^ and 7.92 nm, respectively. As the grafting reaction progresses, the maximum probability pore size and specific surface area of silica gel gradually decrease, as shown in Table 2. After the quaternization reaction, the material still has a large pore size and specific surface area (398.4 m^2^·g^−1^, 6.28 nm), which indicates that the quaternized material SG-VTS-VPQ can be used for adsorption experiments.

### 3.2. Optimal Experimental Results of Graft Polymerization

By investigating the single factors of 4-VP dosage, reaction temperature, and initiator dosage in the reaction of introducing pyridine groups, this part of the experiment determined the optimal grafting conditions of 4-VP and obtained the maximum grafting amount of 4-VP.

Influence of 4-VP dosage on grafting amount:

The change law of grafting amount with 4-VP dosage is shown in Figure 11. As the 4-VP dosage increases, the grafting amount first increases and then decreases. In the range of 1–5 mL, as the 4-VP dosage increases, the grafting amount increases slowly, which is because a higher dosage increases the probability of a collision of 4-VP molecules with C=C on the surface of silica gel, and the amount of grafting increases accordingly. When the 4-VP dosage reaches 6–8 mL, the grafting amount increases significantly, which may be because the polymerization between 4-VP monomers increases rapidly at this concentration, and the molecular weight of the polymer on the surface of the silica gel is significantly improved. When the dosage is greater than 7 mL, due to high concentration, the homopolymer formed by 4-VP is too large to enter the pores of the silica gel and bond with the surface of the silica gel C=C. Therefore, the amount of grafting decreased rapidly.

Influence of reaction temperature on grafting amount:

The effects of reaction temperature on the grafting amount are shown in Figure 12. It can be seen that as temperature rises, the grafting amount shows an upward trend integrally. When the temperature is less than 75 °C, as the temperature rises, the grafting amount increases sharply. This is because when the temperature is low, as the temperature rises, the initiator decomposes to generate free radicals and the polymerization reaction speeds up, which is conducive to the smooth progress of the graft polymerization reaction. As the temperature continues to rise, the promotion effect of this reaction condition approaches saturation, and the change of the grafting amount tends to be stable. When the temperature exceeds 85 °C, the grafting amount is basically stable, about 1.15 mmol·g^−1^.

Influence of initiator dosage on grafting amount:

The effects of initiator on the grafting amount are shown in Figure 13. As initiator dosage increases, the grafting amount first increases and then decreases. When the initiator dosage is less than 60 mg, as the dosage increases, the number of active free radicals generated in the reaction increases, which increases the initiation efficiency of the polymerization reaction and increases the utilization rate of C=C on the surface of silica gel. Therefore, the grafting amount increases gradually. When the initiator continues to increase, too many free radicals (initiator residues) make the probability of chain initiation and chain termination sharply increase, which makes the polymerization reaction terminate quickly and the grafting amount decreases [41].

Orthogonal test:

The results of orthogonal test are shown in Figure 14 and Table 3. By using extreme difference analysis to process the orthogonal results, the primary and secondary relationship of the three influencing factors was C > B > A (initiator dosage > reaction temperature > 4-VP dosage), and the optimal factor combination was A_2_B_3_C_2_, of which 4-VP dosage was 7 mL, reaction temperature was 95 °C and initiator dosage was 60 mg. Under the above optimal preparation process, the grafting amount of 4-VP is 1.25 mmol·g^−1^, as shown in Figure 9b.

### 3.3. Study on the Adsorption Performance of Silicon-Based Quaternized Material for Pu(IV)

Study on the influence of HNO_3_ and NO_3_^−^ concentrations on the adsorption of Pu(IV):

The black trend line in Figure 15 shows the variation of the adsorption capacity of SG-VTS-VPQ on Pu(IV) with the concentration of HNO_3_. When the HNO_3_ concentration is in the range of 1–5 mol·L^−1^, with the increase of acidity, the adsorption capacity of SG-VTS-VPQ to Pu(IV) gradually increases, and the inflection point appears when the concentration reaches 5 mol·L^−1^, which may be caused by changes in NO_3_^−^ or H^+^ concentration. In order to determine the reason, we studied the influence of NO_3_^−^ concentration on the adsorption of Pu(IV) by SG-VTS-VPQ in 1 mol·L^−1^ HNO_3_ system. The experimental results are shown by the red line in Figure 14. As the NO_3_^−^ concentration increases, the adsorption capacity gradually increases, and no inflection point occurs. According to the complexation law of Pu(IV), Pu(IV) can form a series of complexes with NO_3_^−^ from PuNO_3_^3+^ to [Pu(NO_3_)_6_]^2-^, and with the increase of NO_3_^−^ concentration, the stable complex anion [Pu(NO_3_)_6_]^2-^ ratio increased [42]. However, with the increase of H^+^ concentration, Pu(NO_3_)_6_^2-^ has a tendency to form [HPu(NO_3_)_6_]^−^ or H_2_Pu(NO_3_)_6_ [43]. Combining the above rules, we can preliminarily think that the adsorption behavior of SG-VTS-VPQ on Pu(IV) tends to be an anion exchange form.

Figure 16 shows the adsorption changes of Pu(IV) on porous silica gel and activated carbon under different concentrations of HNO_3_. For porous silica gel, the adsorption amount of Pu(IV) did not change significantly with the increase of acidity, so the adsorption of Pu(IV) on the surface of silica gel may not involve reactions such as anion exchange. For activated carbon, the adsorption performance of the material for Pu(IV) remained stable at low concentrations of acid (1–3 mol·L^−1^). However, with the increase of acidity, the adsorption capacity of Pu(IV) gradually decreased; in particular, under the condition of 7 mol L^−1^ HNO_3_, the adsorption capacity decreased rapidly, which may be because the acid resistance of activated carbon is weaker than that of silica gel, and its pore structure destroyed at higher acidity, resulting in decreased adsorption performance. In addition, the adsorption amounts of Pu(IV) on both silica gel and activated carbon are at low levels (about 3.6 mg·g^−1^ for silica gel and about 5.1 mg·g^−1^ for activated carbon), which may be because there are no corresponding active groups on the surface of two materials (only hydroxyl groups on the surface of silica gel, and acid anhydrides, carboxyl groups, etc. on the surface of activated carbon), so only simple physical adsorption can be performed.

Adsorption kinetics study:

The adsorption kinetics result of Pu(IV) on SG-VTS-VPQ was shown in Figure 17. The adsorption amount was increased quickly in the first 70 s, then the growth rate slows down. When it reaches 300 s, the adsorption reaches equilibrium and the adsorption amount remains unchanged after that.

The adsorption kinetics result was analysed by using three adsorption kinetic models, namely pseudo first-order kinetic model (the reaction rate is linearly related to the concentration of a reactant, and this model is based on the fact that rate-determining step is a physical process), pseudo second-order kinetic model (the reaction rate is linearly related to the concentration of two reactants, and this model is based on the fact that rate-determining step is a chemical reaction), and the Elovich kinetic model (the Elovich model is suitable for processes with irregular data or with large activation energy) [44]. The kinetic parameters for the adsorption of Pu(IV) are listed in Table 4. The fitted adsorption capacity (q_e_) was compared with the equilibrium adsorption amount (q_e,exp_) in the experiment. It can be observed from Table 4 that the q_e_ of pseudo first-order kinetic was much closer to the q_e,exp_ compared with other kinetic models. In addition, the determination coefficient (R^2^) of the pseudo first-order kinetic was higher than that of the pseudo second-order and Elovich kinetic model, which indicates that the physical adsorption is the rate-determining step [45]. This may be because that the huge specific surface area of porous silica gel increases the contact area significantly between quaternized groups and the solution and this advantage plays a decisive role in the adsorption process [46].

In the adsorption of Pu(IV), the newly synthesized silicon-based quaternized material SG-VTS-VPQ has better dynamic performance than trioctylmethylammonium chloride (TEVA) resin, which was most commonly used in the past for the adsorption and separation of Pu(IV). In the report of Liu-Bo on “extracting plutonium in water samples by TEVA and analyzing it by liquid scintillation counting method”, the static adsorption of 25 mg·L^−1^ plutonium in aqueous solution was carried out with TEVA resin, and the time to reach the adsorption equilibrium was about 10.3 min, much higher than the material synthesized in this study [47].

Adsorption thermodynamics research:

The parameters and fitted plots of Langmuir and Freundlich adsorption isotherm models are listed in Table 5 and Figure 18. The maximum adsorption capacity of SG-VTS-VPQ for Pu(IV) was 79.96 mg·g^−1^ (0.33 mmol·g^−1^). Based on the higher determination coefficient (R^2^ = 0.9813), the Pu(IV) adsorption by SG-VTS-VPQ was more in line with the Langmuir model, which indicates that the adsorption sites on the adsorbent are homogeneously distributed and adsorption is more inclined to single-layer adsorption [48,49]. Because the important assumption of the Langmuir model is monolayer adsorption, and typical chemical adsorption is also monolayer adsorption, it can be inferred that the adsorption of Pu(IV) by SG-VTS-VPQ may be chemical adsorption. In addition, the Freundlich constant n_f_ was 0.562 (between 0.1 and 1), indicating that the adsorption process of Pu(IV) was favourable [50].

Van’t Hoff’s fitting results and parameters for temperature changes are shown in Figure 19 and Table 6. According to the calculation results of thermodynamic parameters, the ΔH of SG-VTS-VPQ for Pu(IV) adsorption are 27.35 kJ·mol^−1^, which means that the adsorption processes are endothermic. According to the study of Ma et al., the enthalpy change range of physical adsorption is between 2.10 and 20.90 kJ·mol^−1^, and the enthalpy change range of chemical adsorption is between 20.90 and 418.40 kJ·mol^−1^ [51]. In the temperature range of 293.15–333.15 K, the ΔG range of SG-VTS-VPQ for Pu(IV) adsorption is between −4.49 and −0.67 kJ·mol^−1^, namely with the increase of temperature, the ΔG value decreases, indicating that the adsorption process is spontaneous.

Study on adsorption mechanism of Pu(IV) on SG-VTS-VPQ:

In the study of the influence of HNO_3_ and NO_3_^−^ concentration on adsorption, according to the results, we can preliminarily think that the adsorption behavior of SG-VTS-VPQ on Pu(IV) tends to be an anion exchange form. In order to further explore the adsorption mechanism of SG-VTS-VPQ on Pu(IV) in the HNO_3_ system, we conducted the following experiments.

Assuming that the adsorption of Pu(IV) by SG-VTS-VPQ has the following reaction formula:
(12)$$\left(n-4\right){\mathrm{SiPyR}}_{4}^{+}+\mathrm{Pu}{\left({\mathrm{NO}}_{3}\right)}_{n}^{4-n}\overset{K}{\to }\left(n-4\right){\mathrm{SiPyR}}_{4}\cdot \mathrm{Pu}{\left({\mathrm{NO}}_{3}\right)}_{n}$$
(13)$$K=\frac{\left[\left(n-4\right){\mathrm{SiPyR}}_{4}\cdot \mathrm{Pu}{\left({\mathrm{NO}}_{3}\right)}_{n}\right]}{{\left[{\mathrm{SiPyR}}_{4}^{+}\right]}^{n-4}\cdot \left[\mathrm{Pu}{\left({\mathrm{NO}}_{3}\right)}_{n}^{4-n}\right]}=\frac{K_{d}}{{\left[{\mathrm{SiPyR}}_{4}^{+}\right]}^{n-4}}$$
(14)$${\mathrm{lgK}}_{d}=\mathrm{lgK}+\left(n-4\right)\mathrm{lg}\left[{\mathrm{SiPyR}}_{4}^{+}\right]$$
where K is the adsorption equilibrium constant, K_d_ is the distribution ratio, n is the number of complex acid radical ions, [SiPyR_4_^+^] is the content of functional groups (mmol). Take lgK_d_ as the ordinate and lg[SiPyR_4_^+^] as the abscissa to draw a straight line, and the value of n can be obtained after fitting. If *n* = 6, SG-VTS-VPQ only adsorbs Pu(NO_3_)_6_^2−^; 5 < *n* < 6, SG-VTS-VPQ adsorbs Pu(NO_3_)_6_^2−^ and Pu(NO_3_)_5_^−^; *n* = 5, SG-VTS-VPQ only adsorbs Pu(NO_3_)_5_^−^; *n* < 5, the adsorption of Pu(IV) by SG-VTS-VPQ may have other forms besides anion exchange.

The adsorption results of different adsorbent dosages are shown in Table 7, and the fitting results are shown in Figure 20. According to the slope of the straight line, *n* = 5.67, it can be considered that the adsorption of Pu(IV) by SG-VTS-VPQ is mainly complex anions Pu(NO_3_)_6_^2−^ and Pu(NO_3_)_5_^−^.

Separation experiment of trace plutonium in uranium matrix:

Selecting suitable adsorption conditions can improve the adsorption capacity of Pu(IV) on SG-VTS-VPQ, thereby reducing or avoiding the loss of Pu(IV) in the uranium leaching stage and improving the recovery rate of Pu(IV). When [HNO_3_] ≤ 3 mol·L^−1^, UO_2_^2+^ mainly exists in the form of UO_2_(NO_3_)^+^ after complexation with NO_3_^−^, and the form of UO_2_(NO_3_)_3_^−^ is less, so the U(VI) partition coefficient on the material is very low [52]. However, according to the previous static adsorption experiment results, Pu(IV) easily forms Pu(NO_3_)_6_^2-^ in HNO_3_ aqueous solution, and the partition coefficient on SG-VTS-VPQ is very high, so the material can adsorb Pu(IV) in the chromatography column, so as to achieve the separation of uranium and plutonium.

In this work, 2 mol·L^−1^ HNO_3_ and 0.3 mol·L^−1^ HNO_3_-0.3 Na_2_C_2_O_4_ were selected as eluents to elute U(VI) and Pu(IV) in turn. The elution curves of uranium and plutonium were thus obtained on the SG-VTS-VPQ column, as shown in Figure 21. According to the calculation, the uranium content in the uranium eluent is about 96.11% of the total uranium in the original sample, and the recovery rate of Pu(IV) is about 90.68%. The plutonium eluent still contains a small amount of uranium, which is about 0.42% of the total uranium. The decontamination factor DF of uranium in plutonium is 215.9.

Acidity stability test:

The adsorption properties of SG-VTS-VPQ treated with different concentrations of HNO_3_ for Pu(IV) are shown in Figure 22. Figure 22a–f are adsorption results with 1 mol·L^−1^, 3 mol·L^−1^, 5 mol·L^−1^, 7 mol·L^−1^, 8 mol·L^−1^, 9 mol·L^−1^ HNO_3_ treatment, respectively. It can be seen from Figure 22a–c that the adsorption capacity of the silicon-based quaternary ammonium material SG-VTS-VPQ does not decrease with time after soaking in HNO_3_ below 5 mol·L^−1^, indicating that the adsorption performance of the material will not be affected below this acid concentration. When the concentration HNO_3_ reaches 7–8 mol·L^−1^, with the increase of soaking time, the adsorption amount first decreases and then remains unchanged. This is because part of the functional groups of the material are destroyed, and the damaged part may be 4-VP polymer long chain grafted on the surface of silica gel by disproportionation termination. When the acidity reaches 9 mol·L^−1^, the adsorption capacity keeps decreasing with the increase of time, which may be due to the serious damage of the surface functional groups.

According to the change of adsorption performance of the material under different acidity and different soaking time treatment, it can be considered that the silicon-based quaternary ammonium separation material SG-VTS-VPQ will not be damaged below 5 mol·L^−1^; When used in a short-term environment of 7 mol·L^−1^ HNO_3_, the material performance can be basically guaranteed; if it is used in HNO_3_ above 8–9 mol·L^−1^, it will be seriously damaged in a short time.

Figure 23 shows the infrared spectra of SG-VTS-VPQ after soaking with different concentrations of HNO_3_ for 24 h. It can be seen from Figure 23a–c that the C–N stretching vibration peak on the pyridine ring at 1642.09 cm^−1^ and the characteristic peak of quaternary ammonium nitrogen at 1472 cm^−1^ both exist, and after three acidity treatments, the quaternary ammonium nitrogen characteristic peaks’ intensity did not change, indicating that the quaternized functional groups on the surface of SG-VTS-VPQ were not destroyed in HNO_3_ concentrations below 5 mol·L^−1^, which also explained the reason why the adsorption performance remained stable in Figure 22a–c. In Figure 23d, compared with a, b, and c, the C–N stretching vibration peak on the pyridine ring has no change, whereas the intensity of the characteristic peak of quaternary ammonium nitrogen is significantly reduced, indicating that some quaternized functional groups in the material may be damaged and shed. In Figure 23e, due to the significant reduction in the intensity of the characteristic peak of quaternary ammonium nitrogen at 1472 cm^−1^, the originally covered characteristic peak at 1419.29 cm^−1^ (the C–C stretching vibration peak in the pyridine ring) appeared. In Figure 23f, the characteristic peak of quaternary ammonium nitrogen had completely disappeared and were replaced by the pyridine ring stretching vibration peaks at 1429.29 cm^−1^ and 1559.14 cm^−1^, so after 8 mol·L^−1^ and 9 mol·L^−1^ HNO_3_ treatment, most or all of the quaternized groups are removed and the adsorption of SG-VTS-VPQ had a more obvious drop in performance.

Adsorption stability research experiment:

Figure 24 shows the five elution curves of Pu(IV) adsorption–desorption on SG-VTS-VPQ chromatography column. According to the experimental results, the recoveries of Pu(IV) in the five elutions were 96.94%, 94.13%, 94.59%, 90.09%, and 89.81%. Although the overall recovery of Pu(IV) showed a gradual downward trend, the fifth adsorption–desorption recovery could still maintain a high level of about 90%.

## 4. Conclusions

The paper studied the synthesis optimization of the silica-based quaternized adsorption material SG-VTS-VPQ and its adsorption behavior for Pu(IV). By optimizing the synthesis process, the grafting amount of 4-VP reached 1.25 mmol·g^−1^. According to the analysis of NMR and XPS, the quaternization rate of pyridine groups reached 91.13%. In the adsorption experiments, the thermodynamic experiment results show that the adsorption of Pu(IV) by SG-VTS-VPQ is more in line with the Langmuir adsorption model and the adsorption type is a typical chemical adsorption; the kinetic results show that adsorption process is more in line with the pseudo first-order kinetic model, and the larger specific surface area of SG-VTS-VPQ plays an important role in the adsorption; the results of the adsorption mechanism show that the adsorption of Pu(IV) by SG-VTS-VPQ is mainly complex anion Pu(NO_3_)_6_^2−^ and Pu(NO_3_)_5_^−^. This research provides in-depth and detailed ideas for the surface modification and application of porous silica gel, and at the same time provides a new view by which to analyze the development direction of pretreatment materials in the post-treatment process.

## Figures and Tables

**Figure 1 molecules-27-03110-f001:**
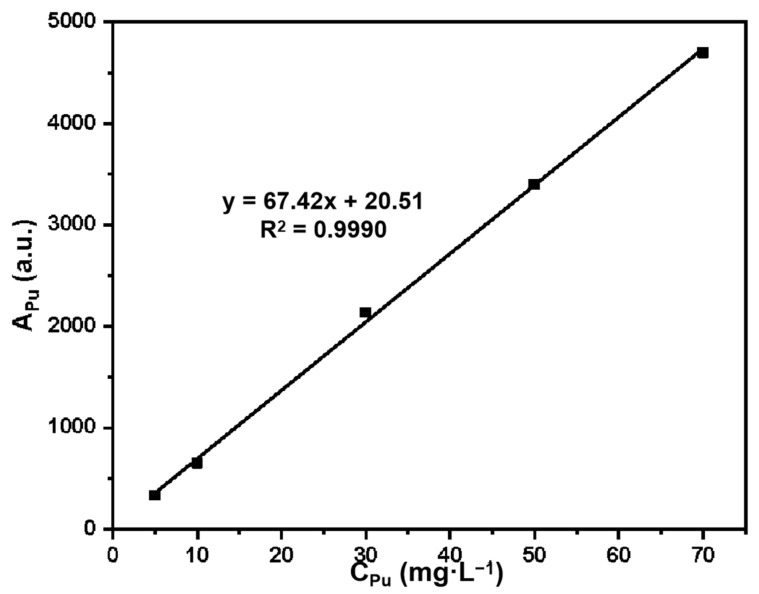
Standard curve for Pu(IV) analysis by X-ray fluorescence spectroscopy.

**Figure 2 molecules-27-03110-f002:**
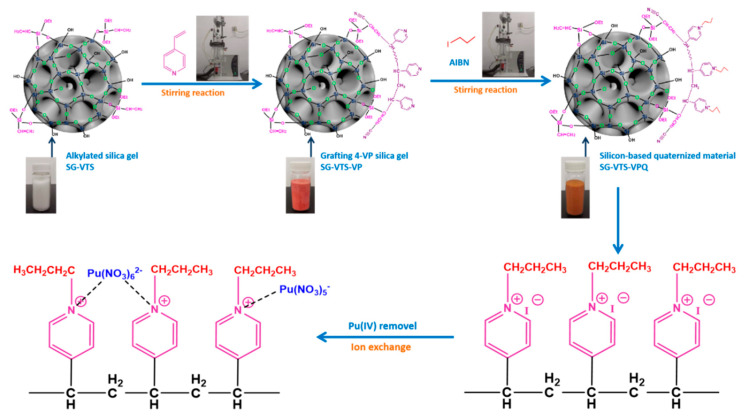
Preparation process of SG-VTS-VPQ and adsorption process of Pu(IV).

**Figure 3 molecules-27-03110-f003:**
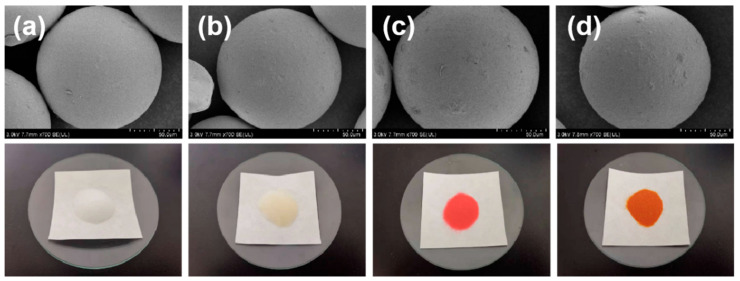
Appearance and SEM images of SG (**a**), SG-VTS (**b**), SG-VTS-VP (**c**) and SG-VTS-VPQ (**d**).

**Figure 4 molecules-27-03110-f004:**
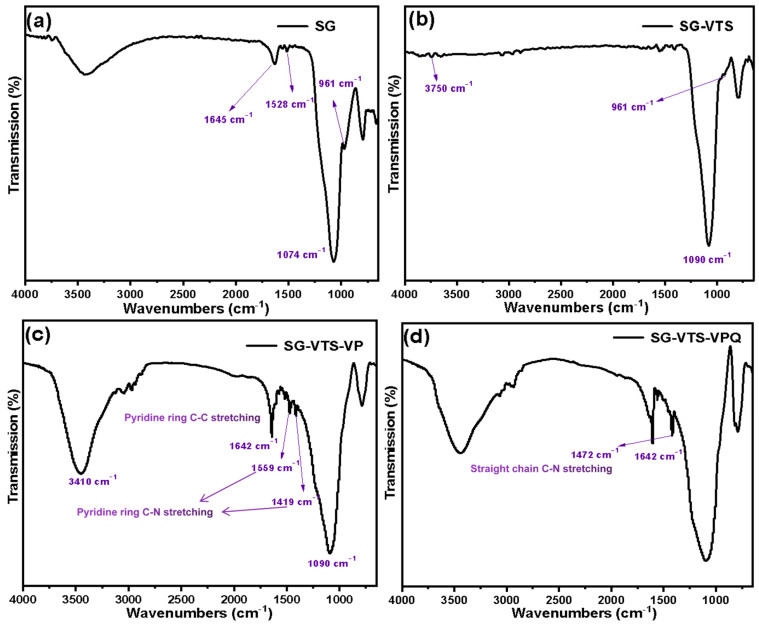
FTIR spectra of the SG (**a**), SG-VTS (**b**), SG-VTS-VP (**c**) and SG-VTS-VPQ (**d**).

**Figure 5 molecules-27-03110-f005:**
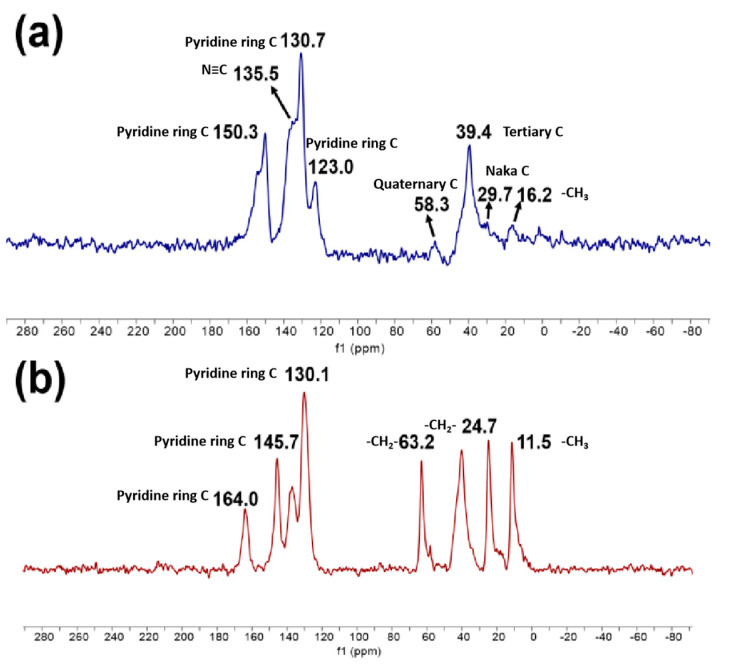
^13^C-NMR spectra of SG-VTS-VP (**a**) and SG-VTS-VPQ (**b**).

**Figure 6 molecules-27-03110-f006:**
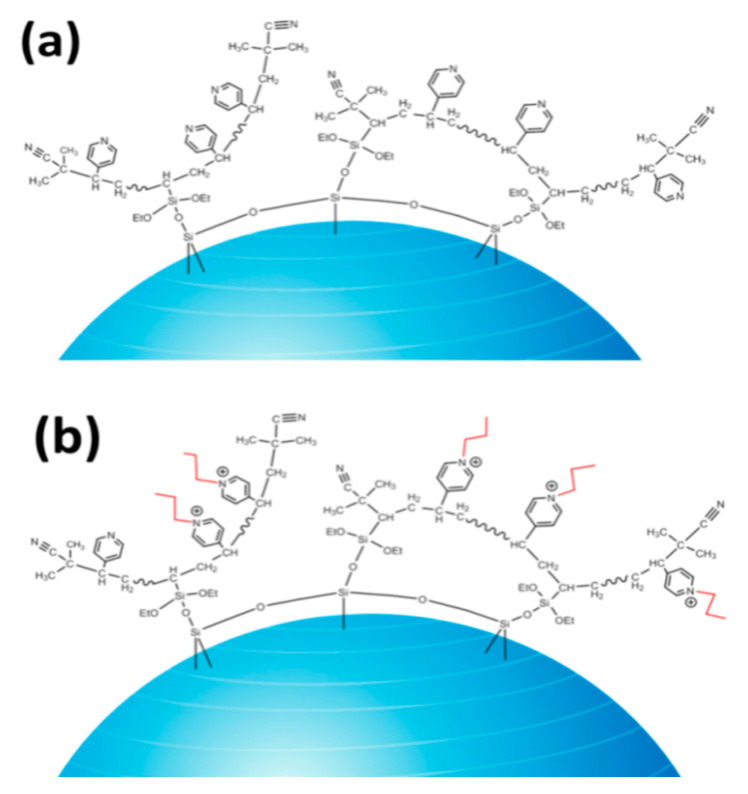
Surface morphology of SG-VTS-VP (**a**) and SG-VTS-VPQ (**b**).

**Figure 7 molecules-27-03110-f007:**
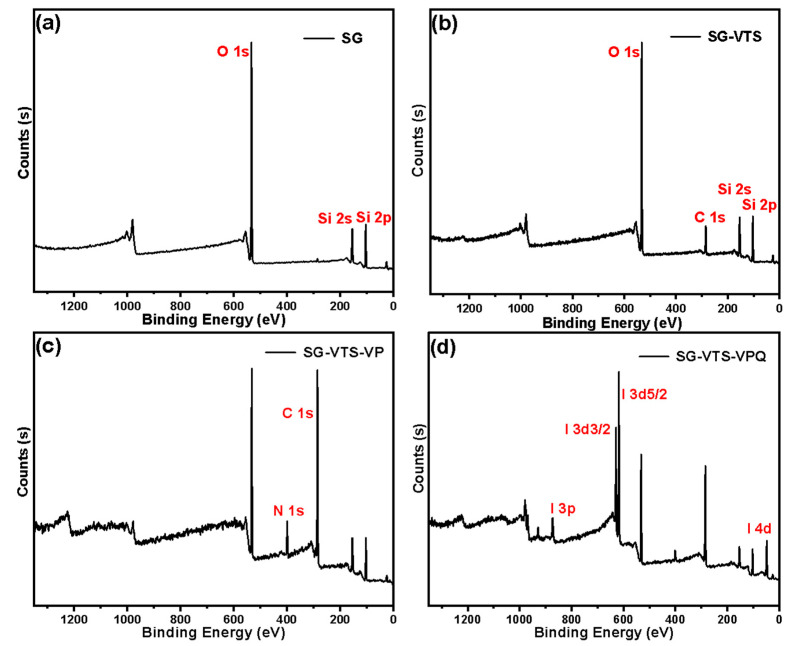
XPS spectra of SG (**a**), SG-VTS (**b**), SG-VTS-VP (**c**), and SG-VTS-VPQ (**d**).

**Figure 8 molecules-27-03110-f008:**
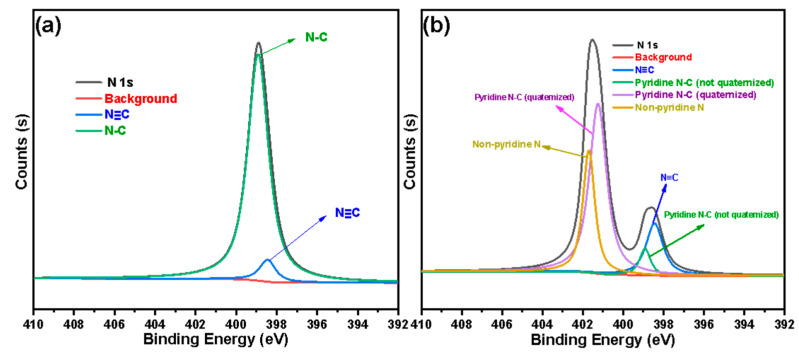
The deconvolution of N 1s spectra: SG-VTS-VP (**a**), SG-VTS-VPQ (**b**).

**Figure 9 molecules-27-03110-f009:**
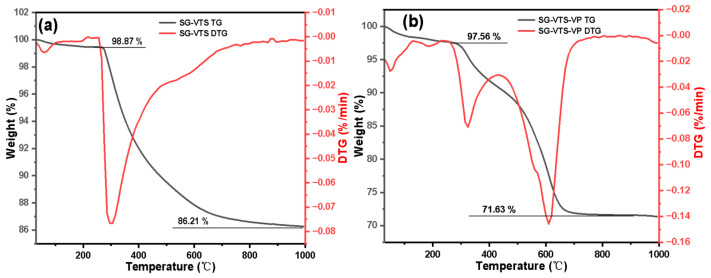
Thermogravimetric curve of SG-VTS (**a**) and SG-VTS-VP (**b**).

**Figure 10 molecules-27-03110-f010:**
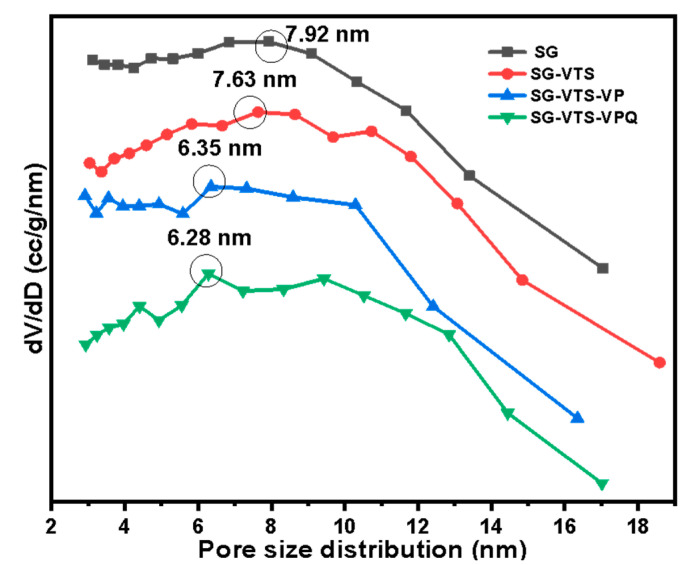
Pore size distribution of SG, SG-VTS, SG-VTS-VP and SG-VTS-VPQ.

**Figure 11 molecules-27-03110-f011:**
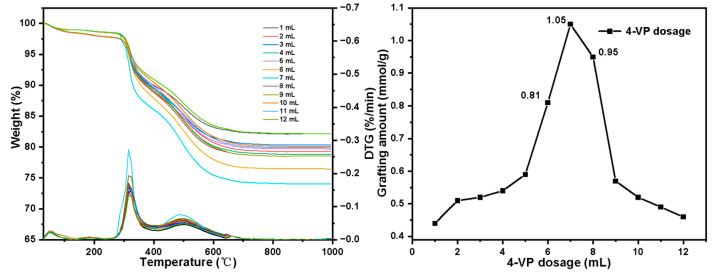
Grafting amount varies with 4-VP dosage change.

**Figure 12 molecules-27-03110-f012:**
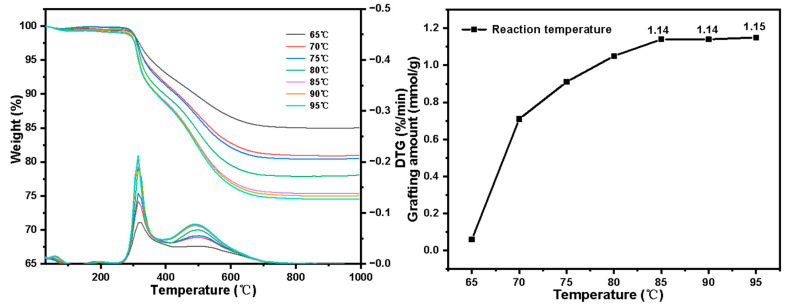
Grafting amount varies with 4-VP reaction temperature.

**Figure 13 molecules-27-03110-f013:**
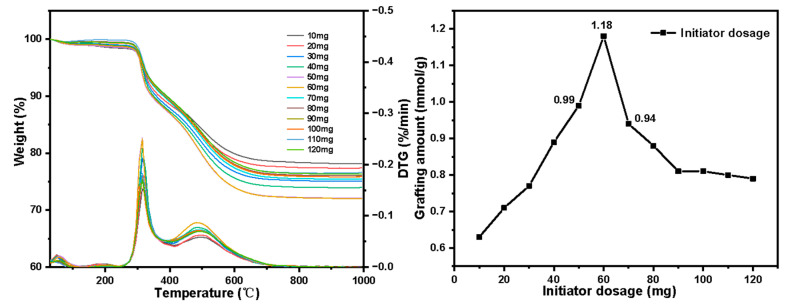
Grafting amount varies with initiator dosage.

**Figure 14 molecules-27-03110-f014:**
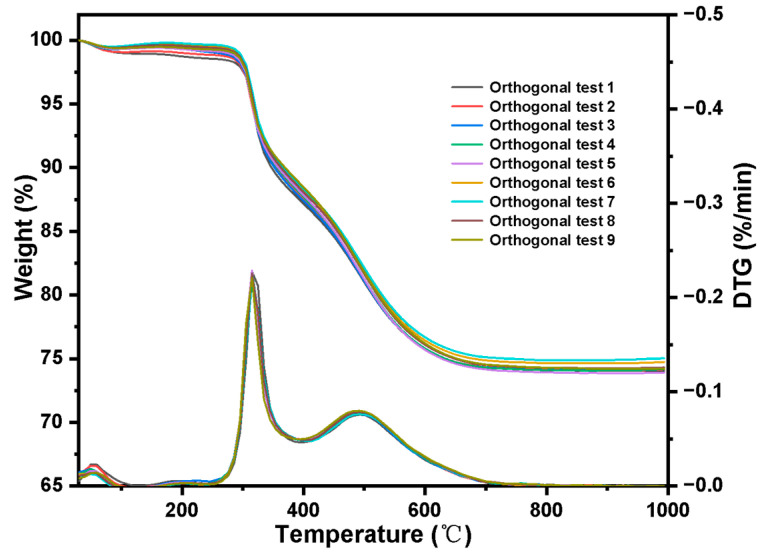
Orthogonal experiment result.

**Figure 15 molecules-27-03110-f015:**
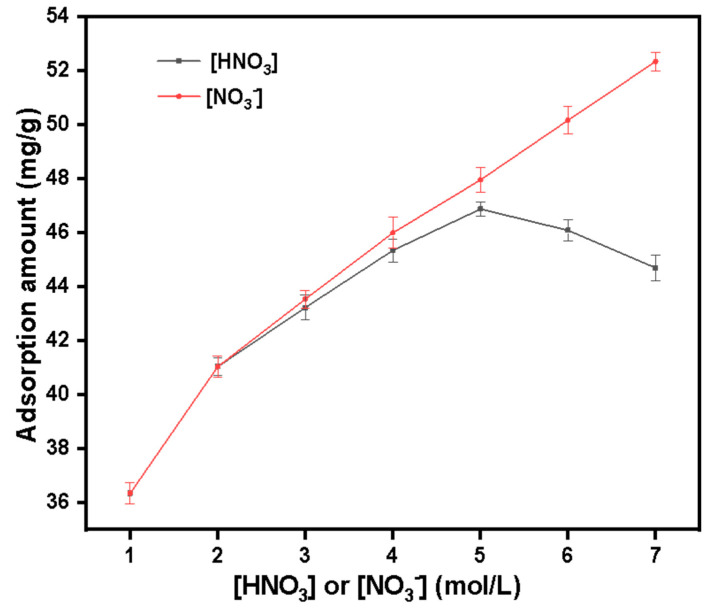
Effect of [HNO_3_] and [NO_3_^−^] on the adsorption of Pu(IV) by SG-VTS-VPQ.

**Figure 16 molecules-27-03110-f016:**
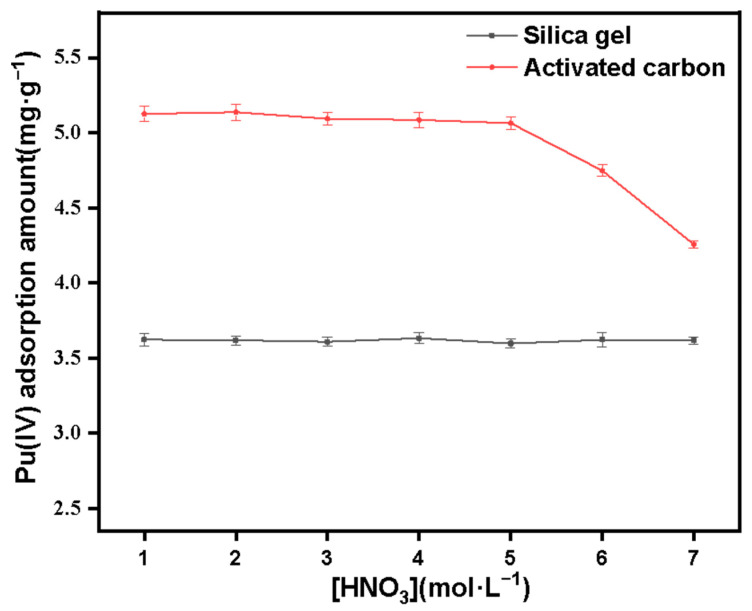
Effect of HNO_3_ concentration on the adsorption of Pu(IV) on silica gel and activated carbon.

**Figure 17 molecules-27-03110-f017:**
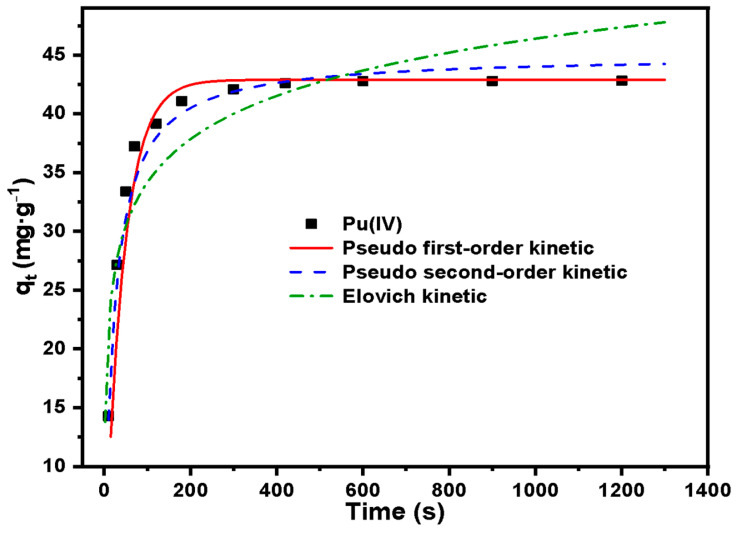
Adsorption kinetic model for Pu(IV) on SG-VTS-VPQ.

**Figure 18 molecules-27-03110-f018:**
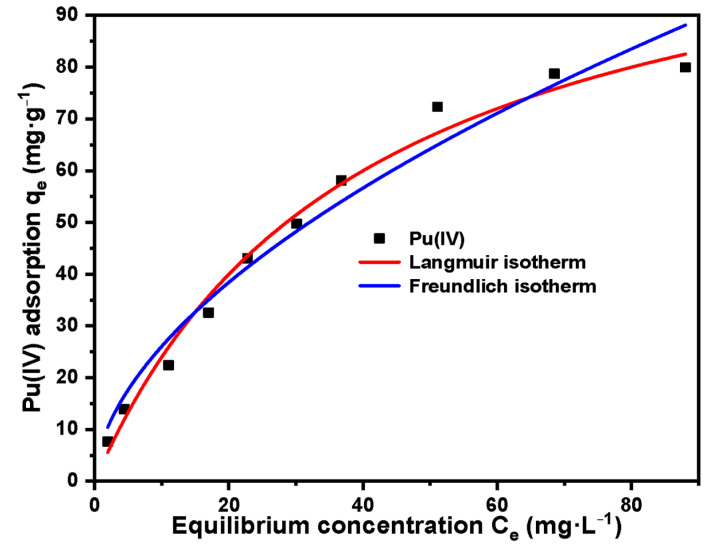
Adsorption isotherm model for Pu(IV) on SG-VTS-VPQ.

**Figure 19 molecules-27-03110-f019:**
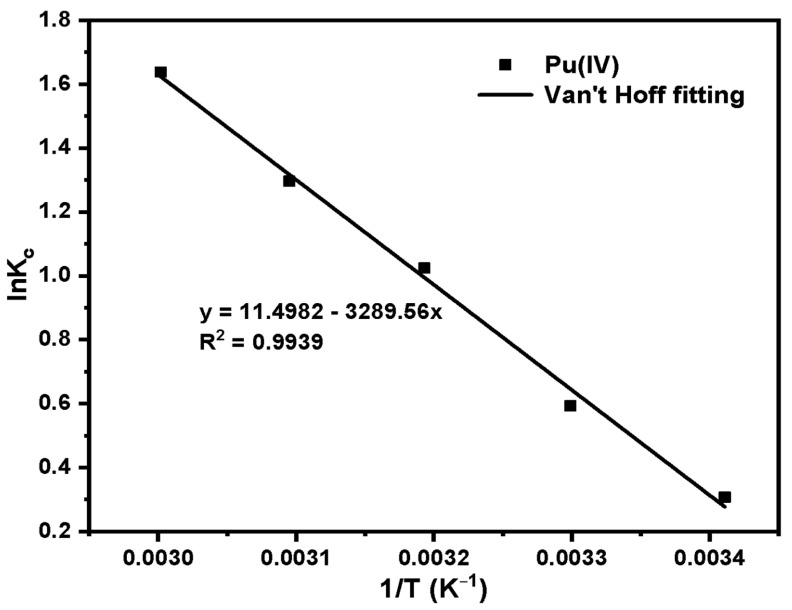
Fitting results of K_c_ and 1/T of Pu(IV).

**Figure 20 molecules-27-03110-f020:**
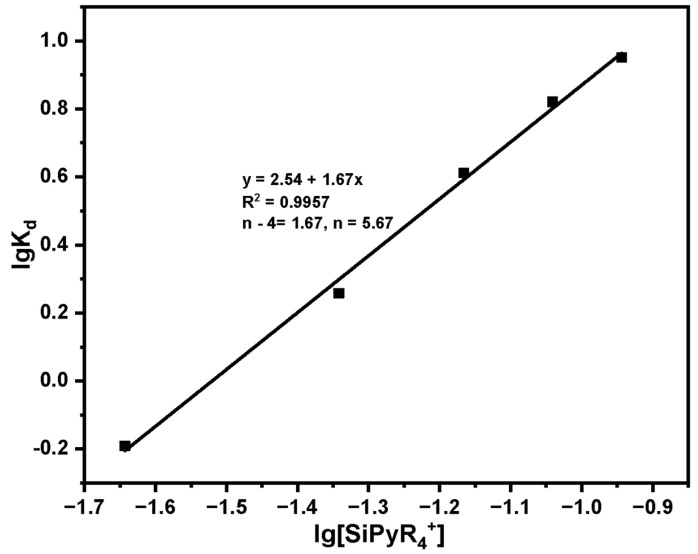
Fitting results of adsorption mechanism.

**Figure 21 molecules-27-03110-f021:**
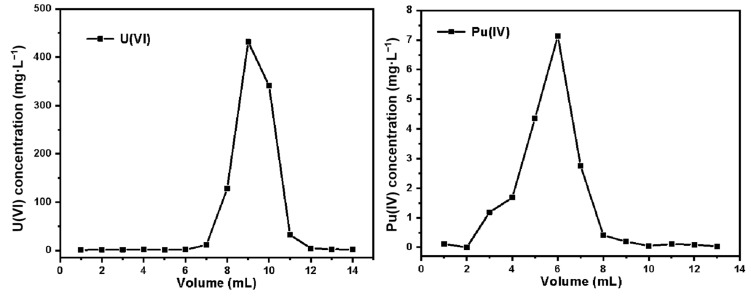
Elution curves for uranium and plutonium.

**Figure 22 molecules-27-03110-f022:**
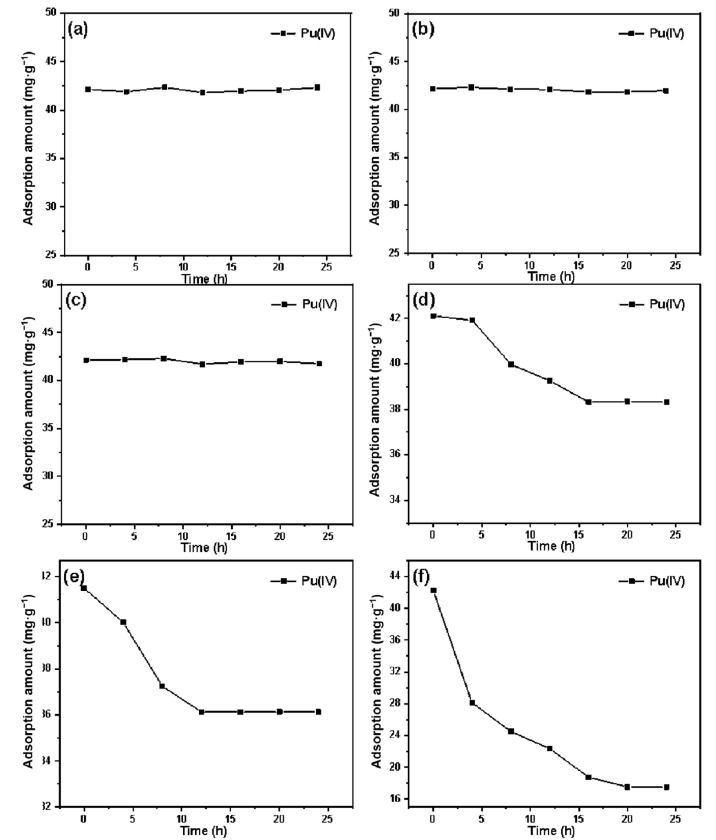
Adsorption of Pu(IV) by SG-VTS-VPQ treated with different concentrations of HNO_3_. (**a**) adsorption of Pu(IV) by SG-VTS-VPQ treated with 1 mol·L^−1^ HNO_3_; (**b**) adsorption of Pu(IV) by SG-VTS-VPQ treated with 3 mol·L^−1^ HNO_3_; (**c**) adsorption of Pu(IV) by SG-VTS-VPQ treated with 5 mol·L^−1^ HNO_3_; (**d**) adsorption of Pu(IV) by SG-VTS-VPQ treated with 7 mol·L^−1^ HNO_3_; (**e**) adsorption of Pu(IV) by SG-VTS-VPQ treated with 8 mol·L^−1^ HNO_3_; (**f**) adsorption of Pu(IV) by SG-VTS-VPQ treated with 9 mol·L^−1^ HNO_3_.

**Figure 23 molecules-27-03110-f023:**
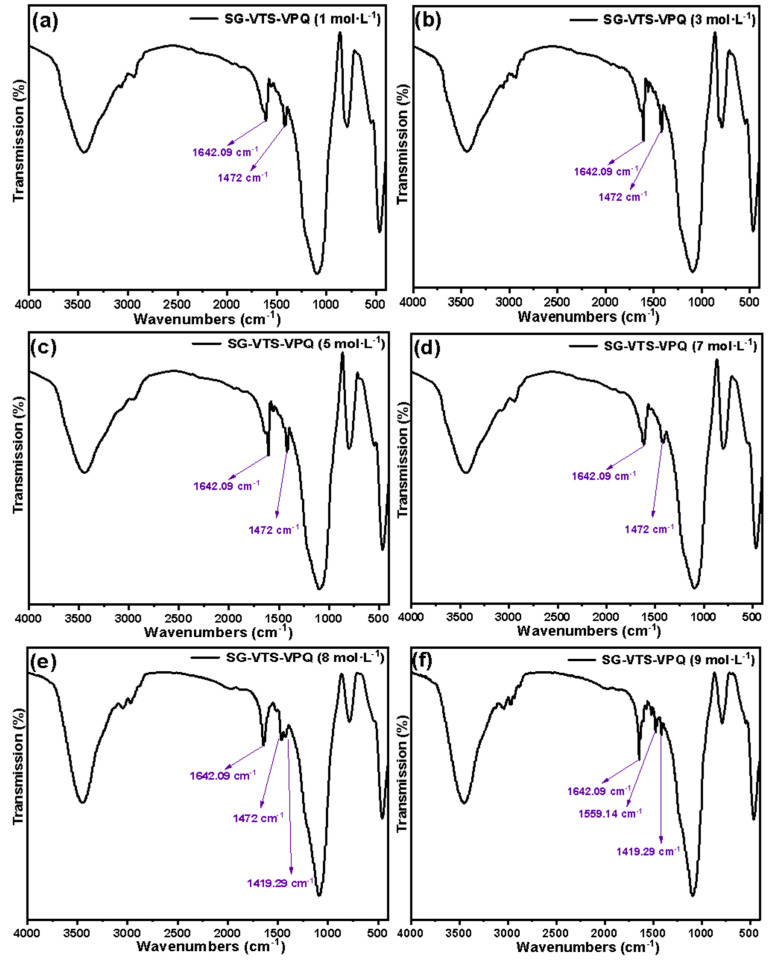
Infrared spectra of SG-VTS-VPQ treated with different concentrations of HNO_3_. (**a**) FTIR of SG-VTS-VPQ treated with 1 mol·L^−1^ HNO_3_; (**b**) FTIR of SG-VTS-VPQ treated with 3 mol·L^−1^ HNO_3_; (**c**) FTIR of SG-VTS-VPQ treated with 5 mol·L^−1^ HNO_3_; (**d**) FTIR of SG-VTS-VPQ treated with 7 mol·L^−1^ HNO_3_; (**e**) FTIR of SG-VTS-VPQ treated with 8 mol·L^−1^ HNO_3_; (**f**) FTIR of SG-VTS-VPQ treated with 9 mol·L^−1^ HNO_3_.

**Figure 24 molecules-27-03110-f024:**
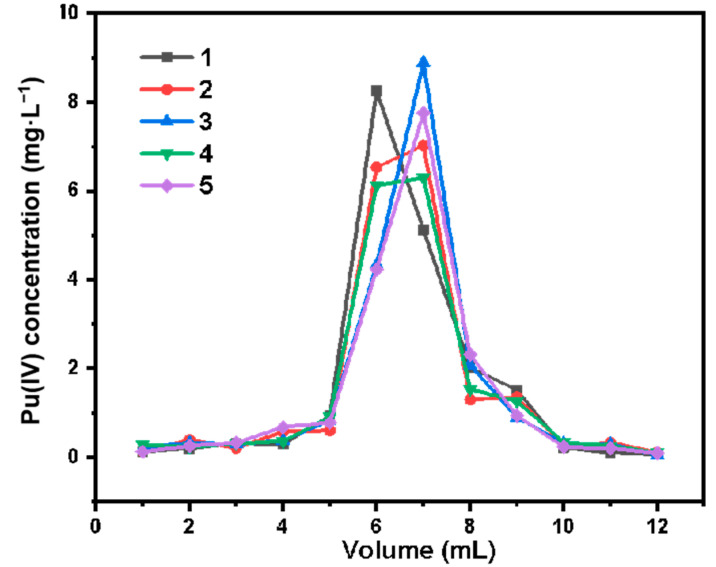
Pu(IV) adsorption-desorption five elution curves on SG-VTS-VPQ column.

**Table 1 molecules-27-03110-t001:** Orthogonal test level distribution.

Factor/Level	1	2	3
4-VP dosage (A)/mL	6	7	8
Reaction temperature (B)/°C	85	90	95
Initiator AIBN dosage (C)/mg	50	60	70

**Table 2 molecules-27-03110-t002:** Changes of specific surface area and pore size with the progress of grafting.

Species	Pore Size (nm)	Specific Surface Area (m^2^·g^−1^)
SG	7.92	540.9
SG-VTS	7.63	503.5
SG-VTS-VP	6.35	427.7
SG-VTS-VPQ	6.28	398.4

**Table 3 molecules-27-03110-t003:** Analysis of orthogonal test results.

	Factors			Grafting Amount (mmol/g)
	4-vp Dosage (mL)	Temperature (°C)	Initiator Dosage (mg)	
1	1	1	1	1.131
2	2	2	2	1.172
3	3	3	3	1.168
4	1	2	3	1.153
5	2	3	1	1.156
6	3	1	2	1.169
7	1	3	2	1.166
8	2	1	3	1.147
9	3	2	1	1.136
K_1_	3.45	3.447	3.423	
K_2_	3.475	3.461	3.507	
K_3_	3.473	3.49	3.468	
K _1_	1.15	1.149	1.141	
K _2_	1.158	1.154	1.169	
K _3_	1.157	1.163	1.156	
R	0.008	0.014	0.028	

**Table 4 molecules-27-03110-t004:** Adsorption kinetic parameters for Pu(IV) on SG-VTS-VPQ.

Adsorption Kinetic Model	Parameters	Pu(IV)
Pseudo first-order	q_e_ (mg·g^−1^)	42.89
	K_1_ (s^−1^)	0.023
	R^2^	0.983
Pseudo second-order	q_e_ (mg·g^−1^)	45.07
	K_2_ (g·s·mg^−1^)	0.001
	R^2^	0.957
Elovich	α_E_ (mg·(g·s)^−1^)	1.483
	β_E_ (g·mg^−1^)	4.313
	R^2^	0.930

**Table 5 molecules-27-03110-t005:** Langmuir and Freundlich adsorption isotherm parameters for Pu(IV) on SG-VTS-VPQ.

Adsorption Isotherm Model	Parameters	Pu(IV)
Langmuir	q_max_ (mg·g^−1^)	119.78
	b (L·mg^−1^)	0.025
	R^2^	0.981
Freundlich	K_f_	7.181
	n_f_	0.562
	R^2^	0.932

**Table 6 molecules-27-03110-t006:** Adsorption thermodynamic parameters of Pu(IV).

Species	T (K)	ΔG (KJ·mol^−1^)	ΔH (KJ·mol^−1^)	ΔS (J·mol^−1^·K^−1^)	R^2^
Pu(IV)	293.15	−4.49	27.35	95.59	0.994
	303.15	−3.54			
	313.15	−2.59			
	323.15	−1.63			
	333.15	−0.67			

**Table 7 molecules-27-03110-t007:** Adsorption mechanism experiment results.

Dosage (mg)	C_e_ (mg·L^−1^)	Q_e_ (mg·g^−1^)	K_d_	lgK_d_	[SiPyR_4_^+^] (mmol)	lg[SiPyR_4_^+^]
20	35.42	22.90	0.646	−0.191	0.0228	−1.643
40	23.23	41.93	1.810	0.258	0.0455	−1.342
60	11.61	47.32	4.074	0.612	0.0683	−1.166
80	6.36	42.05	6.607	0.821	0.0910	−1.041
100	4.12	35.88	8.712	0.951	0.1138	−0.944

## Data Availability

Not applicable.

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
