# Peer review of "Synthesis of Silica-Based Quaternized Adsorption Material and Study on Its Adsorption Behavior for Pu(IV)"

_molecules, 2022, doi:10.3390/molecules27103110_

Round 1

Reviewer 1 Report

The manuscript “Synthesis of silica-based quaternized adsorption material and study on its adsorption behavior for Pu(IV)” proposes silica modified with (N-propyl)-pyridinium iodide, as a material for plutonium extraction for the nuclear wastes processing. It claims possible advantages of the material including its good adsorption capacity and greater radiolysis stability compared to common ion-exchange resins. The manuscript shows a deep investigation of silica modification and reaction optimisation. The plutonium adsorption experiment includes the analysis of sorption kinetics, influence of media (NO3- concentration), thermodynamics and the adsorption mechanism is proposed.

The manuscript is well-written, it presents a deep, accurate and skillful study. No crucial flaws were found, however there are few substantive questions and a lot of secondary remarks which are needed to be cleared before the acceptance of the manuscript. Due to this, the manuscript requires a major revision.

General questions:

  1. Mesoporous silica is known to be a good adsorbent. The authors has found that pores the in modified silica play a substantial role in the plutonium adsorption. However, none data is provided for the plutonium adsorption on unmodified silica. Thus the role of quaternary pyridinium modification on the extraction property is not fully disclosed.
  2. The authors propose the material for the removal of plutonium impurities during nuclear fuel reprocessing. What is the selectivity of the material towards Pu and other elements, e.g. actinides? Please discuss.
  3. One of the advantages of the material is claimed as high stability to radiolysis, however no experiments or discussion is provided.
  4. The silica gel was modified with organic functional groups and then exposed to harsh acidic conditions during plutonium extraction process. What about the stability of prepared material at such conditions? Please comment.

Minor remarks:

    1. All the keywords are used in the Title and the Abstract, so they give no new information to science search engines. The keyword “SG-VTS-VPQ” is strange.
  1. Section: Introduction.
    1. The introduction too terse, it must include more logical goal-setting discussion and underline the importance of the materials for nuclear fuel reprocessing and extraction materials. Please explain the choice of plutonium as a model for extraction experiments.
    2. The acronyms used must be decoded (TEVA, UTEVA, TRU, DGA, SiMaC, SAER, KH-151 etc. etc.…).
  2. Section: Materials and methods.
    1. A commercial silica gel having pore size of 8 nm is described as “macroporous”. It contradicts the IUPAC definition.
    2. The data for the plutonium adsorption was obtain using a self-developed X-ray Fluorescence Analyzer (XFA). No information on the device internals is provided. How was the absolute values of Pu content determined? Please provide at least general description of X-ray fluorescence analyzer, it’s calibration details, data processing, standards being used and typical experimental errors.
    3. In description of SG-VTS-VP optimization batch experiments, only absolute amounts of reagents added are provided. Please provide molar ratios (or their ranges) for convenience.
    4. The description of L9(34) orthogonal experiment needs to be referenced (any standart protocol used?)
  3. Section: Result and Discussion
    1. SEM images are provided at low magnification (x700…x1000), the porous structure can not be visualized. Different magnifications are used for each image which make them hardly comparable.
    2. Please compare plutonium sorption for obtained material to previously reported data.
  4. Some suggestions on data presentation:
  • To improve the readability of NMR (Fig. 3) and FTIR (Fig. 4) please add the peak attributions.
  • Please use the same X-axis interval for both panels in XPS spectra (Fig. 7). Different X-axis tick intervals gives false impression that graphs 7a and 7b depict different Energy loss regions.
  • The changes in the porous structure of silica upon modification (Fig.9) can be improved by combining graphs for different samples into one panel. Five graphs with minor differences are hardly to be compared when separated into different panels.
  • Trim down all the values used to significant digits (BET, XPS, …). There is no sense for providing surface area values to hundredths of square meters or functional groups content up to thundredths of percent.
  1. Section: References
    1. Some references are doubled, e.g. “1. 1 Topin, S…”. Please correct.

Author Response

Dear reviewer:

Thank you for taking the time to review the manuscript and make suggestions to the authors. Based on your questions and suggestions, we responded to the questions point-to-point and revised the manuscript in detail. We have attached replies to your comments in the following article.

Thank you very much for your attention and consideration.

Best regards,

Sincerely yours,

Huibo Li

Reviewer 2 Report

molecules-1699435  Research Paper

Title: Synthesis of silica-based quaternized adsorption material and study on
its adsorption behavior for Pu(IV)

Authors: Wang et al.

General]

The authors synthesized silica-based material for the adsorptive removal of plutonium ions. It is a generally interesting story. Once appropriately revised, the manuscript can be considered for publication in the present journal.   

Specifics]

1] It is recommended not to use abbreviations, acronyms, formulae, and symbols in the manuscript title.  

2] All the experiments should be repeated to ensure data reproducibility. Vertical error bars should be added to each data point accordingly.

3] The authors should conduct an additional reference experiment to show the plutonium adsorption performance of a commercial activated carbon.

4] Can a performance comparison section be added to contrast the performance of the present adsorbent with those reported in the literature for plutonium adsorption?

5] Is the spent adsorbent regenerable? Add the adsorption-desorption cycle data.

Author Response

(The authors gave the same response as above.)

Round 2

Reviewer 1 Report

The authors improved the manuscript greatly. Now, the goal-setting is clear, quaternized silica material is compared to common Pu(IV) adsorption materials. Although the authors did not investigate their material for radiolysis stability, I think that they answered all the crucial questions. I think, the manuscript can now be published.

Just one comment concerning my remark on low magnification SEM images. My own experience shows that silica porous structure can be observed with SEM at high magnifications (x100K – 300K) when low accelerating voltage (e.g., 1-3 kV) are used.